# An N-Cyanoamide Derivative of Lithocholic Acid Co-Operates with Lysophosphatidic Acid to Promote Human Osteoblast (MG63) Differentiation

**DOI:** 10.3390/biom13071113

**Published:** 2023-07-13

**Authors:** Jason P. Mansell, Aya Tanatani, Hiroyuki Kagechika

**Affiliations:** 1School of Applied Sciences, University of the West of England, Coldharbour Lane, Bristol BS16 1QY, UK; 2Department of Chemistry, Faculty of Science, Ochanomizu University, Bunkyo-ku, Tokyo 112-8610, Japan; 3Institute of Biomaterials and Bioengineering, Tokyo Medical and Dental University (TMDU), Chiyoda-ku, Tokyo 101-0062, Japan; kage.chem@tmd.ac.jp

**Keywords:** osteoblasts, differentiation, lithocholic acid, alkaline phosphatase, myocardin-related transcription factor, serum response factor

## Abstract

Less-calcaemic vitamin D receptor (VDR) agonists have the potential to promote osteoblast maturation in a bone regenerative setting. The emergence of lithocholic acid (LCA) as a bona fide VDR agonist holds promise as an adjunct for arthroplasty following reports that it was less calcaemic than calcitriol (1,25D). However, LCA and some earlier derivatives, e.g., LCA acetate, had to be used at much higher concentrations than 1,25D to elicit comparable effects on osteoblasts. However, recent developments have led to the generation of far more potent LCA derivatives that even outperform the efficacy of 1,25D. These new compounds include the cyanoamide derivative, Dcha-150 (also known as AY2-79). In light of this significant development, we sought to ascertain the ability of Dcha-150 to promote human osteoblast maturation by monitoring alkaline phosphatase (ALP) and osteocalcin (OC) expression. The treatment of MG63 cells with Dcha-150 led to the production of OC. When Dcha-150 was co-administered with lysophosphatidic acid (LPA) or an LPA analogue, a synergistic increase in ALP activity occurred, with Dcha-150 showing greater potency compared to 1,25D. We also provide evidence that this synergy is likely attributed to the actions of myocardin-related transcription factor (MRTF)–serum response factor (SRF) gene transcription following LPA-receptor-induced cytoskeletal reorganisation.

## 1. Introduction

It is widely recognised that 1,25-dihydroxy vitamin D3 (1,25D) is essential for the provision of an adequately calcified and mechanically robust bone tissue; 1,25D deficiency in childhood results in rickets, whereas hypovitaminosis D in adults causes osteomalacia [1]. Whilst it is clear that 1,25D ensures the calcification of bone (type I) collagen by mature osteoblasts [2,3], the application of this steroid in a bone regenerative setting is prevented because of its calcaemic action. Hypercalcemia, consequent to prolonged exposure to high 1,25D levels, has devastating consequences that can be fatal. Identifying alternatives to 1,25D that have little or no calcaemic action could be highly desirable for aiding bone repair or promoting the integration of implantable biomaterials such as titanium. One such candidate might be lithocholic acid (LCA) or an LCA derivative following the discovery, by Makishima and colleagues [4], that LCA, a secondary bile acid, is a bona fide agonist of the vitamin D receptor (VDR).

Despite being a VDR agonist, there are reports that LCA is without a significant calcaemic action [5,6]. In light of these studies, we explored the potential of LCA and two LCA derivatives, LCA acetate and LCA acetate methyl ester, to secure human osteoblast maturation [7]. The co-treatment of human (MG63) osteoblasts with LCA and either epidermal growth factor, transforming growth factor beta or lysophosphatidic acid (LPA) resulted in a clear, synergistic maturation response as supported by stark increases in total alkaline phosphatase (ALP) activity, the enzyme responsible for skeletal mineralisation [8]. We also found that LCA could stimulate the activation of activator protein-1, a transcription factor widely recognised to play an important role in the development and maturation of osteoblasts [9,10]. Importantly, compounds targeting either the pregnane X receptor or the farnesoid X receptor could not mimic the action of LCA on osteoblast maturation [7]. Whilst these LCA derivatives generated some very promising findings, we had to use them in the micromolar range (up to 30 μM), whereas 1,25D elicited similar osteoblast maturation responses when used at 100 nM. As part of our ongoing programme to enhance osteoblast maturation on biomaterial surfaces, our search for a more suitable, less-calcaemic 1,25D surrogate continues.

Since our original studies using LCA, the continued interest in this molecule as a surrogate for 1,25D has stimulated the generation of other LCA derivatives with far greater potency. These include the recent amide derivatives [11], especially the N-cyanoamide species, Dcha-150 (Figure 1), which has been found to elicit an HL-60 cell differentiation response with an EC50 of 0.32 nm compared to 0.74 nm for 1,25D. A comparable VDR transactivation was also observed for Dcha-150 in transfected human embryonic kidney (HEK293) cells. Given these very encouraging findings, we wished to examine the ability of Dcha-150 to co-operate with LPA or the LPA analogue, (3S)1-fluoro-3-hydroxy-4-(oleoyloxy)butyl-1-phosphonate (FHBP), in supporting human osteoblast (MG63) maturation. Our focus on LPA/FHBP is informed from our studies reporting on the successful, facile functionalisation of bone biomaterials with this bioactive lipid [12,13,14,15]. Whilst we consistently find a robust maturation response for osteoblasts co-treated with LPA/FHBP and 1,25D/LCA, we are still unclear as to how these stimuli converge in supporting cellular differentiation. To this end, we examined the possible contribution of the MRTF–SRF pathway given its implication in LPA receptor signalling.

## 2. Materials and Methods

### 2.1. General

Unless stated otherwise, all reagents were of analytical grade from Sigma–Aldrich (Poole, UK). Stocks of 1-Oleoyl LPA (Bio-Techne Ltd., Abingdon, Oxford, UK) and FHBP (Tebu-Bio, Peterborough, UK), a phosphatase-resistant LPA analogue, were prepared in 1:1 ethanol:tissue culture grade water to a final concentration of 10 mM and 500 μM, respectively, and stored at −20 °C. Stocks of 1,25D (100 μM) were prepared in ethanol and stored at −20 °C. Dcha-150 was synthesised as detailed previously [11], and stocks (10 mM) were prepared in DMSO and likewise stored at −20 °C. An inhibitor of the myocardin-related transcription factor (MRTF)–serum response factor (SRF) gene transcription pathway, CCG 203971 (Bio-Techne Ltd., Abingdon, Oxford, UK), was prepared in ethanol (5 mM) and also stored at −20 °C. The methods identified below are essentially similar to those reported by us for related works [12,13,14,15].

### 2.2. Human Osteoblasts

Human osteoblast-like cells (MG63) were cultured in conventional tissue culture flasks (250 mL, Greiner, Frickenhausen, Germany) in a humidified atmosphere at 37 °C and 5% CO_2_. Although osteosarcoma-derived [16], MG63 cells exhibit features in common with human osteoblast precursors or poorly differentiated osteoblasts. Specifically, these cells produce type I collagen with no or low basal osteocalcin (OC) and ALP. However, when MG63s are treated with 1,25D, osteocalcin (OC) expression increases, which is a feature of the osteoblast phenotype [17]. When these cells are co-stimulated with 1,25D and selected growth factors, e.g., LPA, there is a synergistic increase in ALP expression [18,19] which reflects cellular maturation. Consequently, the application of these cells to assess the potential pro-maturation effects of selected factors is entirely appropriate. Cells were grown to confluence in Dulbecco’s Modified Eagle Medium (DMEM)/F12 nutrient mix (Gibco, Paisley, Scotland, catalogue number 21331-020) supplemented with sodium pyruvate (1 mM final concentration), L-glutamine (4 mM), streptomycin (20 μg/mL), penicillin (20 units/mL) and 10% *v*/*v* foetal calf serum (Gibco, Paisley, Scotland). The growth media (500 mL final volume) were also supplemented with 5 mL of a 100× stock of non-essential amino acids. Once confluent, MG63s were subsequently dispensed into blank 24-well plates (Greiner, Frickenhausen, Germany). In each case, wells were seeded with 1 mL of 10 kcells/mL suspension (as assessed by haemocytometry). Cells were then cultured for 3 days and the media removed and replaced with serum-free, phenol-red-free DMEM/F12 (SFCM, Gibco, Paisley, Scotland, catalogue number 11039-021) to starve the cells overnight. Osteoblasts were subsequently treated with 1,25D (10 pM–100 nM), Dcha-150 (10 pM–100 nM), LPA (20 μM), FHBP (500 nM) or a combination of these factors in SFCM supplemented with 500 μg/mL fatty-acid-free human serum albumin. Unless stated otherwise, all investigations for Dcha-150 were compared with 1,25D. The medium used for all of the different treatments was SFCM to eliminate any interference with the assays described below. After the desired time point (24–72 h), the conditioned media were processed for OC quantification and the remaining monolayers processed to assess cell number and total ALP activity to ascertain the extent of cellular maturation.

### 2.3. Osteocalcin Quantification in Conditioned Media

The quantification of OC in cell culture media was performed using a proprietary ELISA (Fisher Scientific, Paisley, UK) in accordance with the manufacturer’s instructions. Briefly, samples of media, standards and controls (25 μL) were dispensed into wells already coated with an anti-OC antibody. Once dispensed, each well was treated with 100 μL of an anti-OC antibody conjugated to horse radish peroxidase (HRP) and the plate left to incubate at room temperature for 2 h. Wells were subsequently aspirated and washed three times before treating with 100 μL of HRP substrate. After 30 min, the reaction was terminated and the absorbances read at 450 nm. The data were expressed as the mean pg of OC per 100 k cells ± the standard deviation.

### 2.4. Cell Number

An assessment of cell number was performed using a combination of the tetrazolium compound 3-(4,5-dimethylthiazol-2-yl)-5-(3-carboxymethoxy-phenyl)-2-(4-sulfophenyl)-2H-tetrazolium, inner salt (MTS, Promega, UK) and the electron-coupling reagent phenazine methosulphate (PMS). Each compound was prepared separately in pre-warmed (37 °C) phenol-red-free DMEM/F12, allowed to dissolve and then combined so that 1 mL of a 1 mg/mL solution of PMS was combined to 19 mL of a 2 mg/mL solution of MTS. A stock suspension of MG63s (1 × 10^6^ cells/mL) was serially diluted in SFCM to give a series of known cell concentrations down to 25 × 10^3^ cells/mL. Each sample (0.5 mL in a microcentrifuge tube) was spiked with 0.1 mL of the MTS/PMS reagent mixture and left for 45 min within a tissue culture cabinet. Once incubated, the samples were centrifuged at 900 rpm to pellet the cells, and 0.1 mL of the supernatants were dispensed onto a 96-well microtiter plate and the absorbances read at 492 nm using a multiplate reader. Plotting the absorbances against the known cell number, as assessed initially using haemocytometry, enabled the extrapolation of cell numbers for the experiments described herein.

### 2.5. ALP Activity

An assessment of ALP activity is reliably measured by the generation of p-nitrophenol (p-NP) from p-nitrophenylphosphate (p-NPP) under alkaline conditions. The treatment of cells to quantify ALP activity was similar to that described by us recently [15]. Briefly, the MTS/PMS reagent was removed and the monolayers incubated for a further 5 min in fresh phenol-red-free DMEM/F12 (0.2 mL/well); this was repeated a second time to remove the residual formazan. Following this incubation period, the medium was removed and the monolayers treated with 0.1 mL/well of 7 mM sodium carbonate, 3 mM sodium bicarbonate (pH 10.3) supplemented with 0.1% (*v*/*v*) Triton X-100 to lyse the cells. After 2 min, each well was treated with 0.2 mL of 15 mM p-NPP (di-Tris salt, Sigma, UK) in 70 mM sodium carbonate, 30 mM sodium bicarbonate (pH 10.3), supplemented with 1 mM of MgCl_2_. Lysates were then left under conventional cell culturing conditions for 1 h. After the incubation period, 0.1 mL aliquots were transferred to 96-well microtiter plates, and the absorbance read at 405 nm. An ascending series of p-NP (50–500 μM) prepared in the incubation buffer enabled the quantification of product formation.

### 2.6. F-Actin Visualization

Chamber slides, 4-well (Lab-Tek Chamber Slide systems, Nalge Nunc International, Roskilde, Denmark), were seeded with 1 mL/well of a 5000 cells/mL suspension and the cells left as described above. After the initial culture period, the medium was removed and the cells treated with either SFCM (1 mL) or SFCM supplemented with 500 nM FHBP for 48 h. After the culture period, recovered samples were fixed in 2.5% paraformaldehyde in PBS for 5 min at room temperature. Once fixed, the cells were exposed to a 40,6-Diamidino-2-phenylindole dihydrochloride (DAPI) mountant (Vector Laboratories Ltd., Peterborough, UK) to visualise nuclei and counterstained with Alexa Fluor™ 594 Phalloidin (Fisher Scientific Limited, Loughborough, UK) to detect F-actin. Cells were visualised with a Nikon Eclipse 80i (Nikon Instruments Inc., Melville, NY, USA) fluorescence microscope with a 60× objective and images taken using a DS-Fi1 5 Mega pixel RGB camera (Nikon UK Limited, Kingston upon Thames, UK).

### 2.7. Statistical Analysis

Unless stated otherwise, all experiments described above were performed at least three times, on different days and with different passage numbers of cells. The minimum replicate number, per treatment group, was six. All data were subject to a one-way analysis of variance (ANOVA) to test for statistical significance. Data were deemed to be statistically significant when *p* < 0.05. With the exception of Figure 1, the data provided are representative of three independent experiments.

## 3. Results

### 3.1. Both 1,25D and Dcha-150 Modestly Attenuate Osteoblast Growth

An initial pilot experiment was conducted to ascertain the influence of 1,25D and Dcha-150 (1–100 nM) on MG63 growth in the presence and absence of the mitogen, LPA (20 μM). As anticipated, the treatment of MG63s with LPA for 48 h led to a clear increase in cell number (*p* < 0.0001 compared to the vehicle control) as supported by the greater absorbance at 492 nm (Figure 2). Conversely, the treatment of cells with 1,25D resulted in a modest yet significantly reduced (*p* < 0.001) cellularity at all concentrations tested (Figure 2A). A similar outcome was found for Dcha-150, with each of the concentrations significantly (*p* < 0.001) reducing cell growth (Figure 2B).

### 3.2. Dcha-150 Co-Operates with LPA to Enhance Osteoblast Maturation

Cells from the same pilot experiment were processed for total ALP activity using p-NPP as the substate and quantification of p-NP. The co-stimulation of MG63s with 1,25D (1–100 nM) and LPA (20 μM) led to the expected and significant (*p* < 0.001) dose-dependent increase in p-NP and therefore elevated total ALP activity (Figure 3). A similar effect was observed for cells co-treated with Dcha-150 (1–100 nM) and LPA, although the findings presented demonstrate the greater potency of Dcha-150 at 1 and 10 nM compared to equimolar 1,25D (*p* < 0.0001). The effect of either 100 nM of 1,25D or Dcha-150 in combination with LPA led to a similar, synergistic increase in p-NP.

### 3.3. A Fluoromethylene Analogue of LPA, FHBP, Synergistically Co-Operates with Dcha-150 to Promote Osteoblast Maturation

Having found that LPA and Dcha-150 co-operated to secure MG63 maturation, we examined the effect of FHBP, a potent LPA analogue. Co-treating cells with 10 pM Dcha-150 and 500 nM FHBP for 48 h led to a significant increase in p-NP (*p* < 0.0001) which was comparable to equimolar 1,25D with FHBP (Figure 4). Increasing the concentration of Dcha-150 to 100 pM and 1 nM resulted in further increases in p-NP which were significantly greater than for cells co-stimulated with FHBP and either 100 pM (*p* < 0.0001) or 1 nM 1,25D (*p* = 0.001). Following this initial experiment, a time- and dose-response study was conducted wherein MG63 cells were co-treated with FHBP (500 nM) and either 1,25D (100 pM–10 nM) or Dcha-150 (100 pM–10 nM) for 24, 48 and 72 h (Figure 5). The co-stimulation of cells led to clear increases in ALP activity over time with Dcha-150 yielding the greatest p-NP concentrations for each of the time points and at all doses, compared to 1,25D.

### 3.4. Dcha-150 Stimulates OC Synthesis

As expected, the treatment of MG63s with 1,25D led to the secretion of OC into the conditioned media following a 48 h culture, with a significant increase between 1 and 100 nM (*p* < 0.01). Dcha-150 also promoted OC production, with 1 and 10 nM being significantly more potent (*p* ≤ 0.01) than equimolar 1,25D (Figure 6).

### 3.5. The Inhibition of MRTF–SRF Prevents MG63 Maturation

The application of 5 μM CCG 203971, an inhibitor of the MRTF–SRF gene transcription pathway, effectively prevented the large increase in ALP following FHBP–Dcha-150 co-treatment (Figure 7). The generation of F-actin, in response to LPA receptor signalling, stimulates the MRTF–SRF pathway. Figure 8 provides clear evidence for the accumulation of F-actin in FHBP-stimulated cells.

## 4. Discussion

Less-calcaemic agonists of the VDR have the potential to support bone repair and regeneration, for example, at fracture non-unions and for total joint replacements to bolster their integration into host tissue. The emergence of LCA as a less-calcaemic agonist of the VDR may help realise its application in a bone regenerative setting. However, LCA and some of the earlier derivatives, including LCA acetate and LCA acetate methyl ester, are markedly less potent than 1,25D, a feature that may restrict their clinical use. In our hands, we found that the aforementioned derivatives had to be used in the micromolar range to elicit a comparable 1,25D-induced maturation response for osteoblasts in vitro [7]. This differential effect may be attributed to the finding that these VDR ligands bind to different amino acid residues within the ligand-binding pocket of the VDR. In doing so, each of these agents produces a different VDR ligand complex configuration, which in turn might account for “the compounds’ different biological actions”, possibly via differential effects upon co-activator recruitment and/or co-repressor displacement [20].

Next generation LCA amide derivatives look far more appealing for potential skeletal applications given their greater potency, matching, or even exceeding, their efficacy compared to 1,25D. We therefore turned our attention to a cyanoamide derivative of LCA, Dcha-150, given its compelling pro-differentiating effects on HL-60 cell cultures [11]. Herein, we report that MG63 osteoblasts co-stimulated with Dcha-150 and LPA/FHBP respond by exhibiting a clear and synergistic increase in ALP, a marker of the mature phenotype. Of significant note was the finding that Dcha-150 was more potent than 1,25D in eliciting cellular maturation; a time- and dose-response study revealed that Dcha-150 in combination with FHBP, a potent fluoromethylene analogue of LPA, stimulated MG63 maturation far more effectively than 1,25D for each of the steroid concentrations tested (0.1–10 nM). Promoting an increase in ALP is predicted to ensure the formation of a mechanically competent, mineralised bone matrix; individuals with missense mutations in the ALPL gene have hypophosphatasia, a condition characterised by poorly mineralised bone, sharing similarities with childhood rickets [8]. We also found that Dcha-150 stimulated the synthesis of OC, another marker of differentiating, mature osteoblasts [21]. As with ALP, Dcha-150 was more potent than 1,25D in promoting OC synthesis. Since a vitamin D response element (VDRE) is present within the OC promoter [22], it was highly likely that Dcha-150, similar to 1,25D, would drive OC expression. Conversely, both Dcha-150 and 1,25D did not directly stimulate ALP expression; this is not surprising since there is no functional VDRE for human ALPL, at least for MG63s [23]. Suffice it to say, for 1,25D to stimulate ALP synthesis, it must act in concert with other stimuli, including LPA and selected analogues [24]. The same would appear to be true for Dcha-150.

How LPA/FHBP converges with 1,25D/Dcha-150 to enhance osteoblast ALP is not fully understood, but it is possible that the cytoskeleton is involved in some way. Ridley and colleagues discovered that LPA was the serum-borne factor responsible for driving the generation of F-actin in fibroblasts [25,26,27]. Herein, we show that the treatment of osteoblasts with the LPA analogue, FHBP, results in the accumulation of F-actin, the consequences of which alter cell morphology and intracellular signalling. With regard the latter, it is tempting to speculate that the generation of F-actin from G-actin is connected to the maturation of osteoblasts consequent to co-stimulation with 1,25D/Dcha-150 and LPA/FHBP. G-actin serves as a reservoir for myocardin-related transcription factor (MRTF), a co-activator to the serum response factor (SRF). Importantly, the activity of SRF is dependent on actin polymerisation, following, for example, LPA stimulation [28]. When G-actin monomers polymerise to F-actin upon receipt of LPA/FHBP, MRTF will be released into the cytoplasm. This liberated transcription factor is then able to translocate to the nucleus and influence contractile gene expression upon interaction with SRF [29,30,31,32]. To examine whether MRTF–SRF participates in the expression of ALP for the findings reported herein, we exposed osteoblasts to CCG 203971, an inhibitor of the MRTF–SRF gene transcription pathway [33]. When osteoblasts were treated with the inhibitor, there was no synergistic increase in ALP following Dcha-150–FHBP co-treatment.

These findings support the role of MRTF in the regulation of this enzyme and, potentially, the process of skeletal mineralisation. Credence is given to this possibility following the finding that MRTF influences the balance between the adipogenesis and osteogenesis of human adipose stem cells [34]. Importantly, Hyväri and colleagues found that the inhibition of MRTF led to a reduction in ALP and type I collagen mobilisation for adipose stem cells maintained in an osteogenic medium [34]. Our findings have now shed further light on how LPA/FHBP regulates the expression of ALP in the context of VDR-agonist-stimulated osteoblast differentiation. Further work will need to be conducted to ascertain exactly how MRTF–SRF is linked to the control of ALP expression during the convergence of a pro-mitogenic stimulus (LPA/FHBP) with a pro-differentiating steroid.

## 5. Conclusions

In conclusion, Dcha-150 has the potential to help secure osteoblast maturation where it might be needed most, for example, at or around implantable materials, including total joint prostheses and bone void fillers. The finding that FHBP is more potent than LPA in promoting Dcha-150-induced maturation places this compound as a more desirable adjunct in a bone regenerative setting. There is also scope to consider Dcha-150 for the possible treatment of certain cancers, including osteosarcoma, to help drive cellular differentiation and dampen cell proliferation.

## Figures and Tables

**Figure 1 biomolecules-13-01113-f001:**
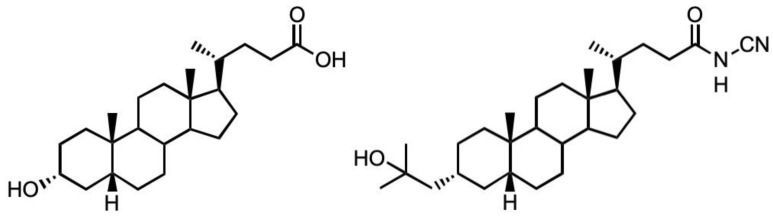
Structures of lithocholic acid (LCA, **left**) and the cyanoamide derivative of LCA (Dcha-150, **right**).

**Figure 2 biomolecules-13-01113-f002:**
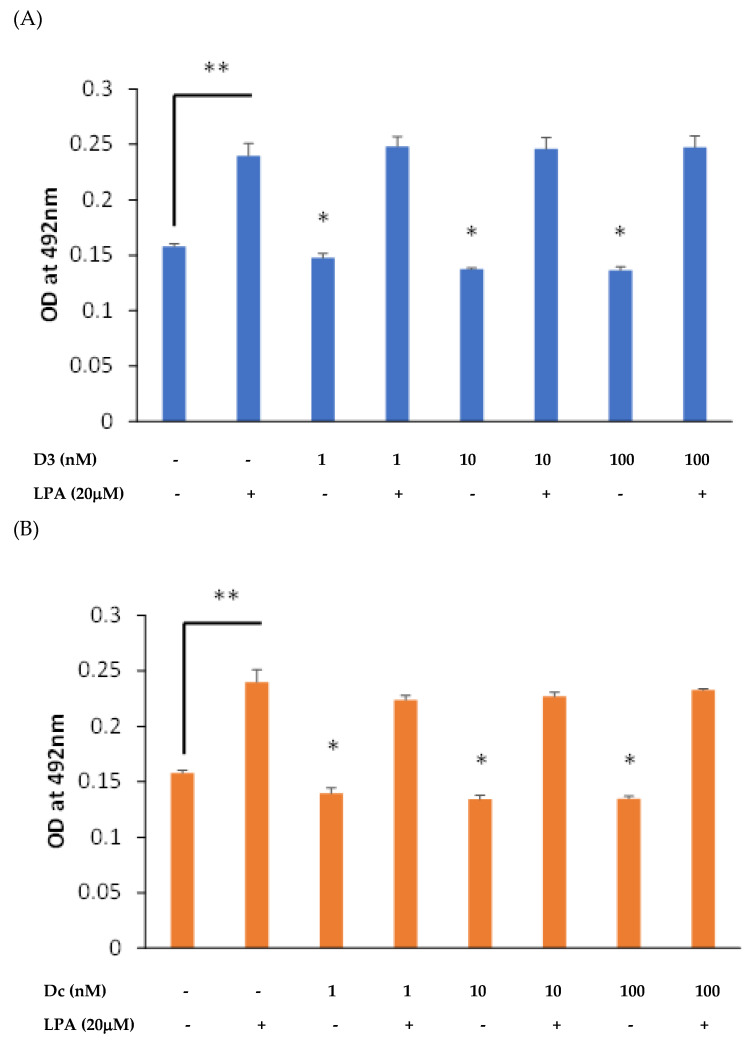
MG63 growth in response to LPA, 1,25D and Dcha-150: (**A**) Osteoblasts were treated with LPA (20 μM), 1,25D (D3, 1–100 nM) or a combination of LPA with 1,25D for 48 h using a serum-free medium. (**B**) Similarly, cells were treated with LPA, Dcha-150 (Dc) or their combination. The assessment of cell growth was via an MTS/PMS assay and absorbance at 492 nm. As anticipated, LPA stimulated MG63 growth (** *p* < 0.01). Compared to unstimulated controls, both 1,25D and Dcha-150 modestly attenuated cell growth at all concentrations tested (* *p* < 0.05). Each bar is the mean from six replicates plus the standard deviation.

**Figure 3 biomolecules-13-01113-f003:**
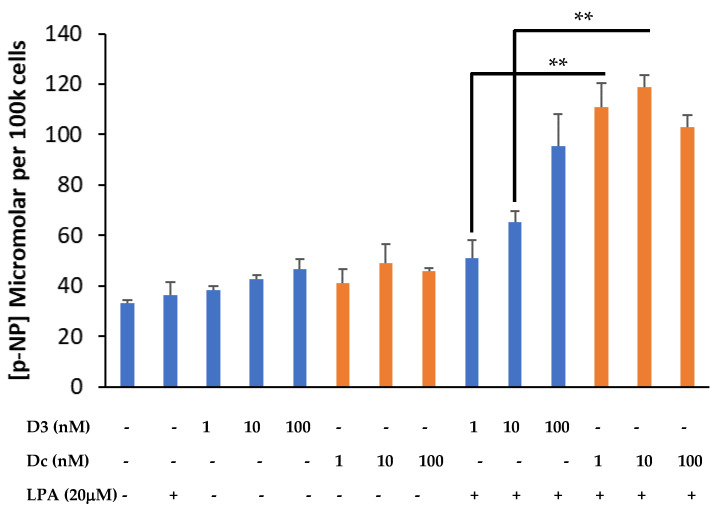
Dcha-150 is more potent than 1,25D in promoting osteoblast maturation. MG63 osteoblasts were treated with LPA (20 μM), 1,25D (D3, 1–100 nM), Dcha-150 (Dc, 1–100 nM) or a combination of LPA with the different VDR agonists for 48 h. The assessment of MG63 maturation was via a total alkaline phosphatase (ALP) assay and quantification of p-nitrophenol (p-NP) from p-nitrophenyl phosphate. The co-stimulation of MG63s with 1,25D and LPA led to increases in ALP activity, which increased with increasing 1,25D concentration. Co-treating osteoblasts with Dcha-150 and LPA generated the greatest increase in ALP activity compared to 1,25D at 1 and 10 nM (** *p* < 0.001). A comparable ALP activity was found for cells co-treated with LPA and 100 nM agonists. Data represent the mean micromolar concentration of p-NP per 100 k cells (6 replicates) + the standard deviation.

**Figure 4 biomolecules-13-01113-f004:**
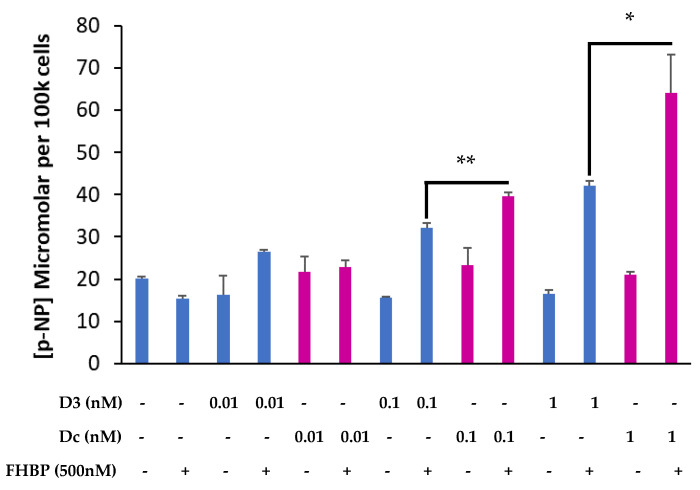
Dcha-150 co-operates with an LPA analogue in promoting osteoblast maturation. Osteoblasts were stimulated with 1,25D (D3, 10 pM–1 nM), Dcha-150 (Dc, 10 pM–1 nM), the LPA analogue (FHBP, 500 nM) or a combination of either steroid with FHBP for 48 h. The assessment of MG63 maturation was via a total alkaline phosphatase (ALP) assay and quantification of p-nitrophenol (p-NP) from p-nitrophenyl phosphate. The co-stimulation of MG63s with FHBP and 100 pM Dcha-150 led to greater ALP activity compared to cells treated with FHBP and equimolar D3 (** *p* < 0.001). Increasing the steroid concentration to 1 nM yielded even greater ALP activity, with Dcha-150 being more potent than 1,25D (* *p* = 0.001). Data represent the mean micromolar concentration of p-NP per 100 k cells (6 replicates) + the standard deviation. The magenta-coloured bars are for Dcha-150-treated cells.

**Figure 5 biomolecules-13-01113-f005:**
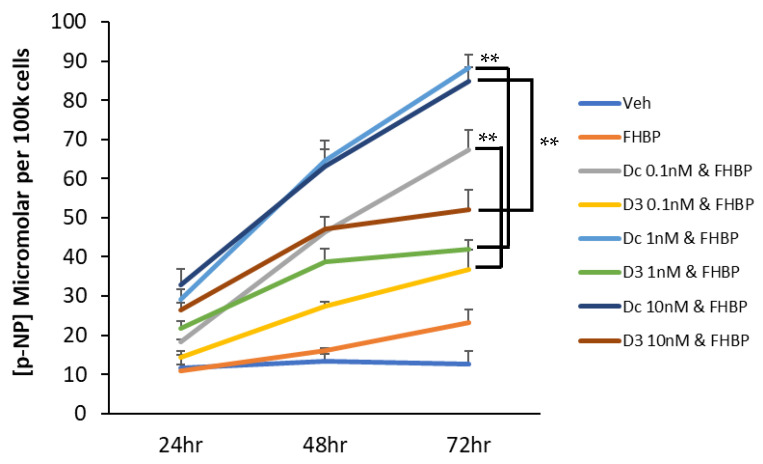
A temporal assessment of osteoblast maturation to co-stimulation with FHBP and Dcha-150—a comparison with 1,25D. Human (MG63) osteoblasts were treated with Dcha-150 (Dc, 100 pM–10 nM), 1,25D (D3, 100 pM–10 nM), FHBP (FH, 500 nM) or FHBP in combination with either VDR agonist for 72 h. After 24, 48 and 72 h of treatment, the cells were processed to ascertain the extent of cellular maturation via total ALP activity. For the sake of figure clarity, the p-nitrophenol (p-NP) data obtained for the steroids used alone (~12 μM per 100 k cells in all cases) have been omitted. The findings depicted clearly demonstrate time and dose-dependent increases in p-NP for co-stimulated cells, with Dcha-150 exhibiting the greatest potency compared to 1,25D. After 72 h, the p-NP for Dcha-150 and FHBP co-treated cells was significantly greater for all steroid concentrations compared to 1,25D (** *p* < 0.001). Each time point is the mean micromolar concentration of p-NP per 100 k cells (6 replicates) + the standard deviation.

**Figure 6 biomolecules-13-01113-f006:**
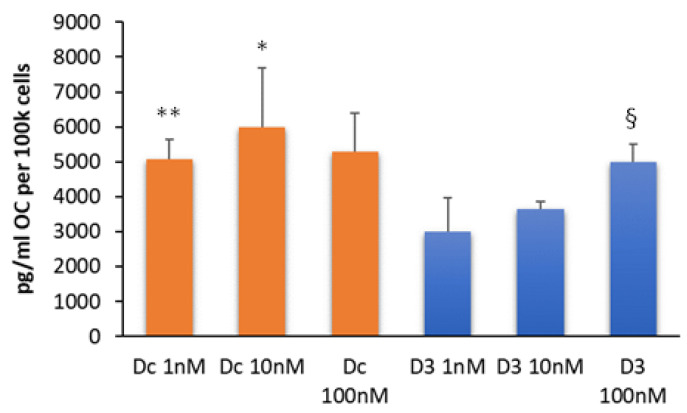
Dcha-150 stimulates osteocalcin synthesis. MG63 cells were treated with Dcha-150 (Dc, 1–100 nM) for comparison with 1,25D (D3, 1–100 nM) on their ability to promote OC synthesis after a 48 h culture. As anticipated, there was evidence of OC secretion into the conditioned media for MG63 cells treated with 1,25D, which increased significantly (^§^
*p* < 0.01) between 1 and 100 nM. The lithocholate derivative, Dcha-150, also stimulated OC synthesis. There were no statistically significant differences between each of the groups. However, 1 and 10 nM Dcha-150 were more potent than equimolar 1,25D in stimulating OC synthesis (** *p* < 0.01, * *p* = 0.01, respectively). OC was undetectable in vehicle-treated cells. Each bar is the mean from six replicates plus the standard deviation.

**Figure 7 biomolecules-13-01113-f007:**
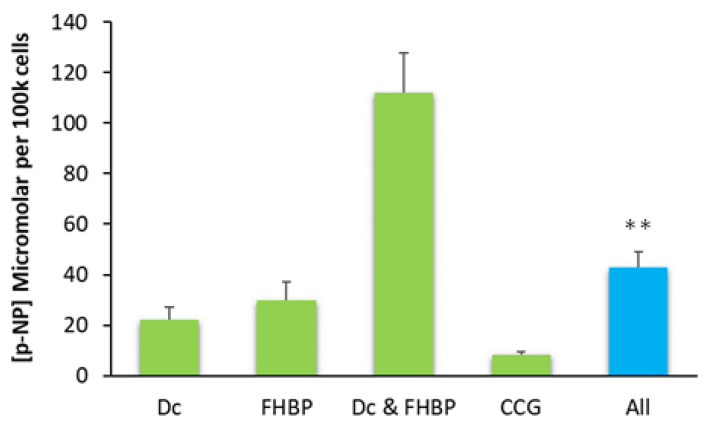
Evidence for the role of the myocardin-related transcription factor (MRTF)–serum response factor (SRF) pathway in the maturation of MG63 cells treated with Dcha-150 and FHBP. MG63s were co-stimulated with FHBP (500 nM) and Dcha-150 (Dc, 1 nM, as informed from Figure 5) for 72 h in the presence (“All”) and absence of CCG 203971 (CCG, 5 μM), an inhibitor of MRTF–SRF gene transcription. The clear and significant inhibition of ALP activity for cells exposed to all agents (** *p* < 0.001 compared to FHBP and Dc co-treated cells) supports the role of MRTF–SRF in the maturation of osteoblasts to FHBP–Dcha-150 co-stimulation. The data are the mean plus the standard deviation from two pooled, independent experiments, each performed on a different occasion using a different passage number of cells (*n* = 8 for Dc, *n* = 10 for all other groups).

**Figure 8 biomolecules-13-01113-f008:**
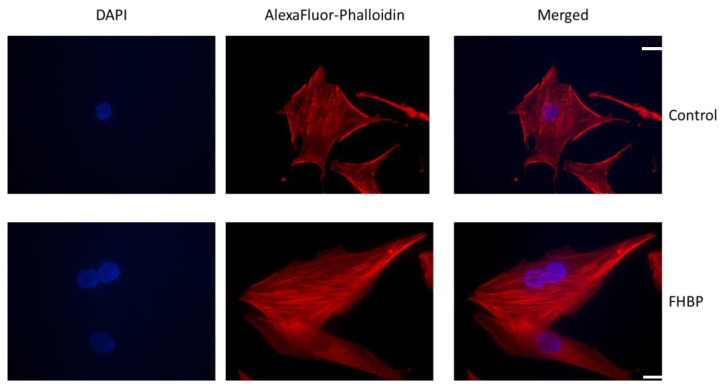
The LPA analogue, FHBP, stimulates the accumulation of F-actin. MG63 cells were seeded into 4-well chamber slides and either treated with serum-free medium alone (control) or 500 nM FHBP for 48 h. Cells were subsequently processed to visualise both nuclei (DAPI) and F-actin (AlexaFluor-Phalloidin). Scale bar: 20 μM.

## Data Availability

The datasets generated for this study are available from J.P.M. on reasonable request.

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
