# Peer review of "An N-Cyanoamide Derivative of Lithocholic Acid Co-Operates with Lysophosphatidic Acid to Promote Human Osteoblast (MG63) Differentiation"

_biomolecules, 2023, doi:10.3390/biom13071113_

Round 1

Reviewer 1 Report

It is an interesting paper that introduced the use of lithocholic acid derivative together with lysophosphatidic acid analog.

However, the hypothesis should be more explicitly stated. The authors can consider using primary human osteoblast cells if this is for translational. Besides osteocalcin, the authors should evaluate calcium deposition as well because it is one the last differentiation markers

Just minor adjustments.

Reviewer 2 Report

This study comes as part of the long ongoing research by the authors:

-         - Lysophosphatidic acid (LPA) cooperation with 1a,25(OH)2D3 (refs. 18-2006; ref. 24-2014)

-         -  Lithocholate (ref. 7-2009)

-          - Biomaterials functionalized with Fluorophosphonate or analogues (Refs. 12-2016; ref. 13-2018; ref. 14-2020; ref. 15-2020)

-          .- Interplay 1a,25-dihydroxy vitamin D3/Cytoskeletal reorganization in MG63 osteoblast maturation (ref. 19-2009)

The present study analysed the effects of D3, LPA, Dc-150, FHBP, isolated or combined, in the maturation of MG63 osteoblast cells, using a similar methodology. Considering the previous information, the reported results, although original, are expected.

My main concern is about the methodology used for ALP activity (Section 2.5). According to the protocol, authors evaluated the levels of ALP, but not the ALP activity. For the activity, levels of ALP must be normalized (to total protein content of the cell layer / number of the cells of the cell layer, …). Apparently, data presented in Figs. 3, 4, 5 are ALP levels, not ALP activity, as the levels were not normalized. If this is the case, it is a relevant issue. Data presented in Fig. 2, regarding cell viability/proliferation (MTS/PMS; OD at 492nm) show significantly increased proliferation in the same conditions that also show higher ALP levels – so, ALP activity (normalized levels) in these conditions would be much lower than that presented in Figs. 3, 4, 5.

Regarding the same issue, for the FHPB effects (isolated or combined), only ALP levels were presented (Figs. 4, 5) – but data on cell viability/proliferation (MTS/PMS) should be also shown – it is expectable that higher ALP levels would be related, at least partially, to a higher number of cells. Thus ALP levels must be normalized to the cell content (total protein / cell number, ..) in the same cell layer. 

-    In the Conclusion, authors only focused on the effect of Dcha-150. It is not clear to me why the authors did not emphasize also the much higher effect when it is combined with FHPB.

-      Also, in the title “An N-cyanoamide derivative of lithocholic acid co-operates with lysophosphatidic acid to promote human osteoblast (MG63) differentiation”, mention to FHPB did not appear.

Why FHPB was apparently forgotten in the Title and Conclusion?

Reviewer 3 Report

In this paper, Mansell et al. investigated the effect of Dcha-150, an amide derivative of lithocolic acid, on human osteoblasts using the MG63 cell line. The authors showed that Dcha-150 exhibits a synergistic effect on osteoblastic differentiation when combined with a mitogen. Furthermore, they observed that the effect of Dcha-150 was inhibited by CCG203971, an inhibitor of MRTF-SRF. I have a couple of questions and comments regarding this paper.

1. Through this study, the ranges of Vitamin D3 and Dcha-150 doses were not constant. For example, in figures 2 and 3, the authors used doses of 1, 10, and 100 nM of Dcha-150. However, in figure 4, they used doses of 0.01, 0.1, and 1 nM of Dcha-150. Then, in figure 5, they used doses of 0.1, 1, and 10 nM, but in figure 6, they used doses of 1, 10, and 100 nM of Dcha-150. Why did the authors reduce the dose of Dcha-150 by two magnitudes in figure 4 compared to figures 2 and 3? Why did they increase the dose of Dcha-150 by one magnitude in figure 5 compared to figure 4? Furthermore, why did they further increase the doses of Dcha-150 by one magnitude in figure 6? All these inconsistencies in the doses have confused me and made it harder to determine appropriate doses. Moreover, I did not find any explanation regarding why the authors chose a single point dose (1 nM) of Dcha-150 in figure 7.

2. Regarding the experiment in figure 7, it was unexpected that the authors introduced the MRTF-SRF pathway without prior mention. The authors discussed the background, rationale, and implications of this pathway in the Discussion section (lines 332-394). However, it would be more appropriate to include the background and rationale in the introduction and/or results sections. Furthermore, the sentence in line 335-337 is confusing, as if the authors published reference #19. However, it appears that they did not. Therefore, the authors should revise this sentence accordingly. Lastly, it is essential for the authors to present their experimental results regarding the accumulation of F-actin in osteoblasts.

3. In figure 2, it is not clear where the single asterisks were compared. The authors should label clearly in the graphs.

Reading this paper was a little uncomfortable. I suggest asking English editing service to improve readability of the manuscript.

Round 2

Reviewer 2 Report

Regarding Comment 1, i agree with the explanation given by the authors. However, ALP levels must always be normalized. I think the authors could do this easily for data on Fig.3, normalizing by the number of cells (i think it might be easy to get the number of cells from the OD values on Fig.2, although not entirely correct) - The Fig.3 graph for normalized ALP levels in the tested conditions would have a much higher visual impact.

For data on Fig. 4 and Fig.5, authors did not give data for the cell proliferation/cell numbers - and this would be essential to get the integrated effect of the compounds - further allowing to normalize ALP levels. Believe me, besides being the correct way to express ALP activity, it will give a higher visual impact.

I also find Fig. 5 not visually friendly for the reader. I understand the need of this type of graph to put together all data, but the reader has to look/analyse the graph several times to get the message. The visual impact of the message from a graph should be immediate.

Reviewer 3 Report

The authors answered all my concerns. I'm satisfied.

Author Response

Thank you for your review of my manuscript.